

# Movement velocity in the chair squat is associated with measures of functional capacity and cognition in elderly people at low risk of fall

Carlos Balsalobre-Fernández[1,2,*], Ángel Cordón[3,*], Nazaret Unquiles[3,*] and Daniel Muñoz-García[4,*]

[1] Department of Physical Education, Sport and Human Movement, Universidad Autónoma de Madrid, Madrid, Spain
[2] LFE Research Group, Department of Health and Human Performance, Faculty of Physical Activity and Sport Science-INEF, Universidad Politécnica de Madrid, Madrid, Spain
[3] Departamento de Fisioterapia, Universidad Autónoma de Madrid, Centro Superior de Estudios Universitarios La Salle, Madrid, Spain
[4] Motion in Brains Research Group, Instituto de Neurociencias y Ciencias del Movimiento, Universidad Autónoma de Madrid, Centro Superior de Estudios Universitarios La Salle, Madrid, Spain
* These authors contributed equally to this work.

Corresponding author
Daniel Muñoz-García,
danimgsan@gmail.com

## ABSTRACT

**Background:** The purpose of this study was to analyze the relationships between muscular performance consisting of a single repetition on the chair squat exercise (CSQ) and different measures of functional capacity, balance, quality of life and cognitive status in older adults.

**Methods:** A total of 40 participants (22 women, 18 men; age = 72.2 ± 4.9 years) joined the investigation. Muscular performance was assessed by measuring movement velocity in the CSQ with no external load using a validated smartphone application (*PowerLift* for iOS). Functional capacity, balance, quality of life and cognitive status were evaluated using the hand-grip strength (HGS) test, the Berg-scale, the EuroQol 5D (EQ-5D) and the Mini mental state examination questionnaire (MMSE). Finally, participants were divided into two subgroups ($N = 20$) according to their velocity in the CSQ exercise.

**Results:** Positive correlations were obtained between movement velocity in the CSQ and HGS ($r = 0.76$, $p < 0.001$), the Berg-scale ($r = 0.65$, $p < 0.001$), the EQ-5D ($r = 0.34$, $p = 0.03$) and the MMSE ($r = 0.36$, $p = 0.02$). Participants in the fastest subgroup showed very likely higher scores in the Berg-scale (ES = 1.15) and the HGS (ES = 1.79), as well as likely higher scores in the MMSE scale (ES = 0.69).

**Discussion:** These results could have potential clinical relevance as they support the use of a time-efficient, non-fatiguing test of muscular performance (i.e., the CSQ) to evaluate functional capacity and mental cognition in older adults.

## INTRODUCTION

Aging can produce a remarkable decrement on muscle mass, exercise performance and bone mineral density, along many other health-related variables (*Evans & Lexell, 1995*; *Ruiz et al., 2008*; *Allison et al., 2013*). Specifically, muscular performance has been associated with the risk of suffering falls, which is a strong factor that leads to morbidity or mortality in the elderly (*Granacher, Zahner & Gollhofer, 2008*; *Volaklis, Halle & Meisinger, 2015*). Resistance training has shown to be a very effective intervention to increase functional capacity, i.e., the ability to perform daily life tasks such as standing up from a chair or walking with proper balance (*Granacher, Zahner & Gollhofer, 2008*; *Granacher et al., 2013*; *Marques, Izquierdo & Pereira, 2013*). Specifically, the increases in muscle strength and power following a resistance training program are believed to be one of the most important factors that prevent falls and reduces frailty in the elderly (*Pijnappels et al., 2008*; *Granacher et al., 2013*; *Lopez et al., 2017*). Moreover, studies have even shown that progressive resistance training can improve cognitive function in older adults with mild cognitive impairment (*Liu-Ambrose, 2010*; *Gates et al., 2013*; *Mavros et al., 2017*).

Several investigations have used different testing protocols to objectively quantify the balance, functional and mental capacities in older adults (like the hand-grip strength (HGS) test, the Berg-scale, the chair test or the EuroQol questionnaire, EQ-5D) in order to develop strategies to prevent falls, reduce frailty and improve the quality of life (*Pijnappels et al., 2008*; *Churchward-Venne et al., 2015*). More recently, smartphone apps have shown to provide accurate measures for different health and performance-related variables in comparison with more expensive laboratory equipment (*Bort-Roig et al., 2014*; *Balsalobre-Fernández, Glaister & Lockey, 2015*; *Balsalobre-Fernández et al., 2018*). For example, the *PowerLift* app for iOS devices was shown to provide valid and reliable measures of movement velocity in the squat exercise in comparison with linear transducers (absolute bias = $-0.005 \pm 0.04$ m/s; standard error of estimate (SEE) = 0.04 m/s, intraclass correlation coefficient (ICC) > 0.9) (*Balsalobre-Fernández et al., 2017*). Finally, during the last decade, the measurement of movement velocity in tasks like the squat has been proposed as a reliable method to quantify muscular performance(*Conceição et al., 2016*; *Muñoz-López et al., 2017*; *García-Ramos et al., 2017a*; *Balsalobre-Fernández, García-Ramos & Jiménez-Reyes, 2017*); in this sense, it is known that, for a certain mass, higher movement velocity results in greater force and power production (*Samozino et al., 2008*; *Muñoz-López et al., 2017*). Taking into account that muscular performance has been proven to be related with functional capacity and mental cognition (*Pereira et al., 2012*; *Marques, Izquierdo & Pereira, 2013*; *Mavros et al., 2017*; *Lopez et al., 2017*) and considering that studies have shown that mean velocity is a reliable measure of muscular performance (*Bazuelo-Ruiz et al., 2015*; *Muñoz-López et al., 2017*; *García-Ramos et al., 2017b*), it is reasonable to think that the analysis of movement velocity during functional tasks in elder populations (such as standing up from a chair) could have relevant clinical applications. However, to the best of our knowledge, no studies have analyzed movement velocity in tasks like the chair squat (from now on, CSQ) in older adults, nor have investigated its potential associations with measures of balance, functional capacity and cognition.

For this, the aim of this study was to analyze if mean task velocity of a single repetition on the CSQ using a smartphone app is related to other widespread balance, functional and cognitive tests in older adults at low risk of falling. Our hypothesis is that velocity in the CSQ will be related with several measures of strength, balance and cognition.

## MATERIALS AND METHODS

### Experimental approach to the problem

This investigation is a correlational study with parallel groups comparison. Performance in functional capacity tests as well as scores in cognition and mental questionnaires were correlated with mean velocity in the CSQ task. Participants were also divided into two groups based on their performance in the CSQ test for comparison purposes.

### Participants

Forty participants from the same nursing home who met the eligibility criteria were recruited for this study ($n = 40$; 22 women, 18 men; age = 72.2 ± 4.9 years; body mass index = 27.8 ± 3.3 kg/m$^2$). Eligibility criteria were established as follow: (1) age between 65–85 years old; (2) not having joined any resistance training program at least in the past five years; (3) not having any diagnosed musculoskeletal disorder or injury; (4) being categorized at low risk of falling in the *Berg* scale. The Spanish version of the Yale Physical Activity (Y-PAQ) Questionnaire (*Katz et al., 2014*) was administered to evaluate participant's basal levels of physical activity, who scored 52.7 ± 23.2 points. According to this questionnaire, participants with a score of 51 points or less are categorized as *sedentary.*

The study protocol complied with the Declaration of Helsinki for Human Experimentation and was approved by the ethics committee at the institutional review board (CSEULS-PI-059/2015). Written informed consent was obtained from each subject before participation.

### Procedures

During a single morning of testing conducted in their nursing home (approximately, from 10 am to 2 pm), participants performed several tasks to test their balance and functional capacities. No warm-up was conducted prior to these tasks. First, the HGS (in kilograms) was tested with a hydraulic dynamometer (Lafayette Instrument Evaluation, Sagamore Parkway North Lafayette, IN, USA) as reported in the literature: participants hold the dynamometer with the wrist in a neutral position and the elbow fully extended (*Pereira et al., 2012*). If the participant supinated their wrist or flexed their elbow, the measurement was repeated. Two attempts were conducted with each hand, and the final score was calculated as the average value of the best attempt of each hand. Second, the Berg-scale was calculated as the sum of scores of 14 different balance and functional tasks as described elsewhere (*Berg et al., 1992*; *Fernandez-Alonso, Muñoz-García & Touche, 2016*). Scores in the Berg-scale were categorized as follows: high risk of falling (0–20), moderate risk (21–40) and low risk (>41) (*Fernandez-Alonso, Muñoz-García & Touche, 2016*). Third, the Y-PAQ (*Katz et al., 2014*), the EuroQol (EQ-5D component) and the

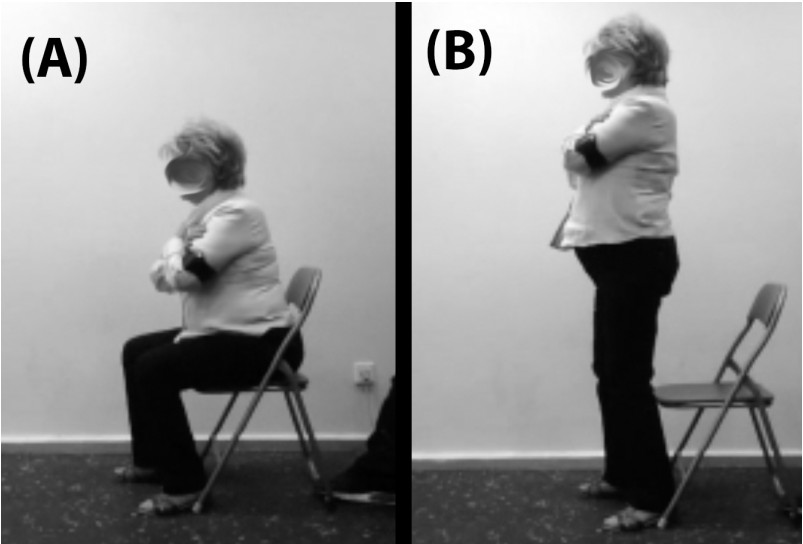

**Figure 1 Initial (A) and final (B) position during the chair squat exercise (CSQ).**

Mini mental state examination (MMSE) (*Cano Gutiérrez et al., 2016*) questionnaires were administrated to evaluate levels of daily physical activity, quality of life and cognitive status, respectively.

Finally, we asked the participants to perform a CSQ: a modification of the squat exercise in which the task starts with the participant sat in a chair. We instructed the participants to sit on a chair (which was the same for every participant) without touching its backrest with their back, to place their feet flat on the ground and at the width of their hips, to keep their arms crossed in their chest and to stand up as fast as possible until the knees were fully extended, and the trunk was in an upright position. Experienced researchers and certified strength and conditioning specialists carefully observed every attempt and asked the participants to repeat the CSQ in case the aforementioned criteria for a valid attempt were not met. See Fig. 1 for more details. We measured mean velocity (in meter per second) of four attempts (separated by 10 s of passive rest) using a validated app (PowerLift for iOS, v.5.4) (*Balsalobre-Fernández et al., 2018*) installed on an iPhone 6 running iOS 10.3.3 (Apple Inc., Cupertino, CA, USA). To measure mean velocity, *PowerLift* uses the well-known Newtonian Eq. (1):

$$v = \frac{d}{t} \tag{1}$$

where $v$ is the mean velocity (in meter per second), $d$ (in meters) the range of motion of the movement (in this case, the difference between the height of the participants standing and sitting on a chair) and $t$ the time (in seconds) of the task, which was calculated by the app as the time between two frames selected by the user. The beginning of the task was considered as the first frame in which the participant rose from the chair, and the end of the task was considered as the first frame in which the knees were completely extended. To record the videos, a researcher held the iPhone on his hand in

portrait position and recorded each lift from the side of the participant at 1.5 m from the chair in order to see the full range of motion as close as possible. The height of the participants, both in the standing and sitting position, was measured using a wall-mounted stadiometer (Seca, Hamburg, Germany). The best of the four attempts was used for the statistical analyses.

We divided the participants in two groups based on their performance in the CSQ. G1 included the half who reached higher velocities in the test, while G2 included the other half of participants.

## Statistical analyses

We used the Pearson's product-moment correlation coefficient to calculate the association between movement velocity in the CSQ and the functional measures. The level of significance was set at $p < 0.05$. To compare groups, we used standardized mean differences (SMD) with the corresponding 90% confidence interval as proposed by *Hopkins et al. (2009)*. The criteria for interpreting the magnitude of the SMD were: trivial (<0.2), small (0.2–0.6), moderate (0.6–1.2), large (1.2–2.0), and extremely large (>2.0). Quantitative chances of better or worse scores were assessed qualitatively as follows: <1%, almost certainly not; 1–5%, very unlikely; 5–25%, unlikely; 25–75%, possible; 75–95%, likely; 95–99%, very likely; and >99%, almost certain. If the changes of better or worse were both >5%, the true difference was assessed as unclear.

## RESULTS

### Correlation between variables

The mean velocity in the CSQ was positively and significantly correlated with the HGS ($r = 0.76$, $p = 0.00$), the Berg-scale ($r = 0.65$, $p = 0.00$), the Y-PAQ ($r = 0.34$, $p = 0.03$), the EQ-5D ($r = 0.34$, $p = 0.03$) and the MMSE scale ($r = 0.36$, $p = 0.02$). See Fig. 2 for more details.

### Difference between groups

Participants in G1 showed very likely higher scores in the Berg-scale (99/1/0; SMD = 1.15, 90% CI [0.51–1.18]) and the HGS (100/0/0; SMD = 1.79, 90% CI [0.98–2.59]), as well as likely higher scores in the MMSE scale (87/10/3; SMD = 0.69, 90% CI [−0.04–1.42], $p < 0.05$) and the Y-PAQ (93/5/2; SMD = 0.87, 90% CI [0.13–1.61], $p < 0.01$). No other variable showed meaningful, statistically significant differences between G1 and G2. See Fig. 3 for more details.

## DISCUSSION

The main finding of this study was the moderate to large positive correlations observed between mean velocity in the CSQ and both the hand grip strength and the Berg-scale. These results are in line with previous research that showed remarkable associations between the 30 s chair test (i.e., completing as many CSQ repetitions as possible within 30 s) and other measures of functional capacity (*Churchward-Venne et al., 2015*; *Bongers et al., 2015*). In this sense, several studies have showed that increased levels of

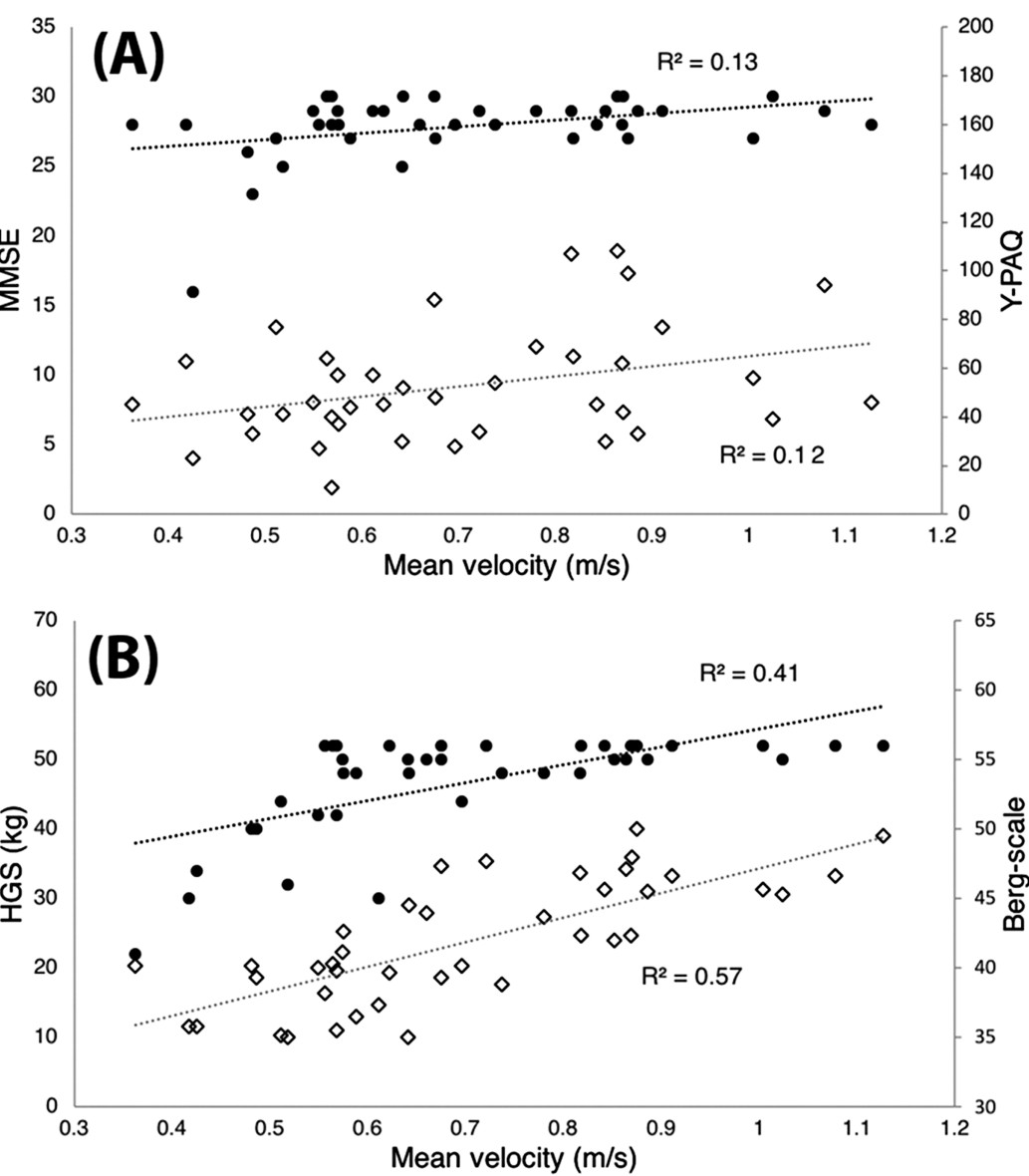

**Figure 2 Mean velocity.** Correlation between mean velocity (in meter per second) in the chair squat exercise and (A) the Mini mental state examination test (MMSE, black dots) and the Yale Physical Activity Questionnaire (Y-PAQ, white diamonds), and (B) the hand grip strength (HGS, black dots) and the Berg-scale (white diamonds).

lower-limbs muscular performance (which traditionally has been tested measuring the times that the participant could sit and stand on a chair in 30 s) are strongly associated with lower risk of falls (*Pijnappels et al., 2008*; *Churchward-Venne et al., 2015*; *Lopez et al., 2017*). However, the CSQ test described in our study could be a proposed as a faster, less fatiguing test for the measurement for older adults since it only requires a single repetition.

We observed moderate to large differences between groups for both the hand grip strength and the Berg-scale, with participants in the strong group (G1) having likely to

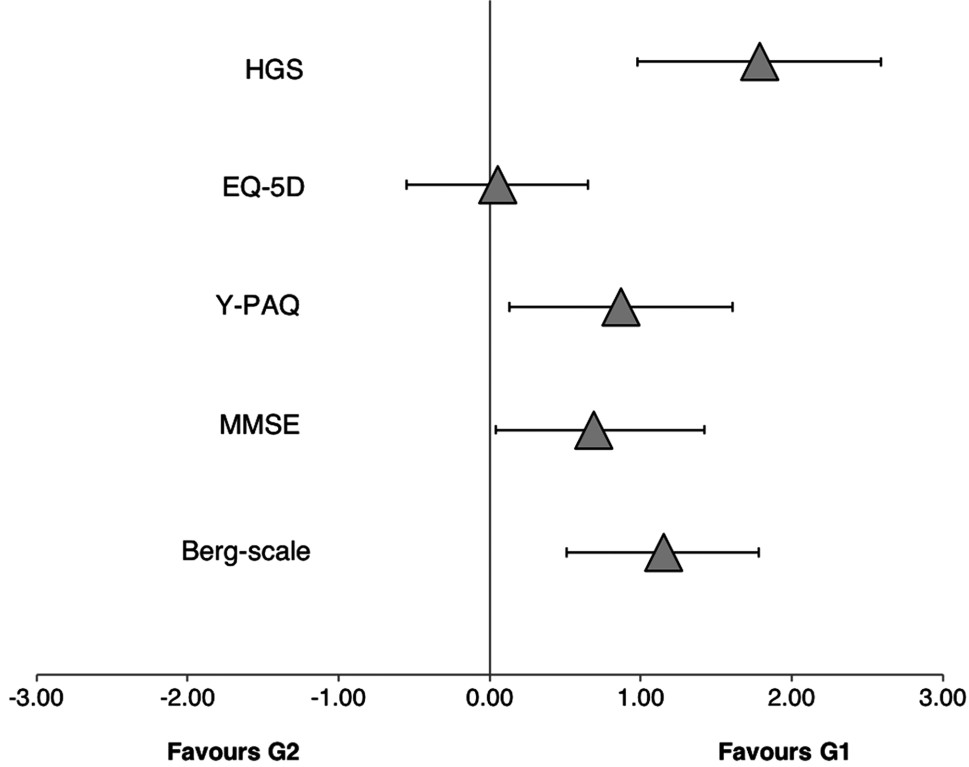

**Figure 3 Standardized mean differences between scores.** Standardized mean differences (with 90% CI) between scores for the hand grip strength (HGS), the EQ-5D scale, the Yale Physical Activity Questionnaire (Y-PAQ), the Mini mental state examination (MMSE) and the Berg scale from the strongest (G1) and weaker (G2) participants.

very likely higher scores than their counterparts from G2. Muscular strength seems to be an extremely important factor to prevent falls and increase functional capacity in the elderly, and recent studies have even shown that there is an inverse relationship between strength and mortality (*Pijnappels et al., 2008*; *Granacher et al., 2013*; *Volaklis, Halle & Meisinger, 2015*; *Stamatakis et al., 2017*; *Lopez et al., 2017*). During the last decade, movement velocity in resistance exercises (such as the back squat) has been proposed to be a very reliable metric to monitor muscular strength since, for a certain load, higher movement velocity means having higher maximal strength and power (*Gonzalez-Badillo & Sánchez-Medina, 2010*; *Muñoz-López et al., 2017*; *García-Ramos et al., 2017b*). In fact, it has been demonstrated that high-speed power training is an effective exercise approach leading to large gains in muscle performance and functional capacity (*Pereira et al., 2012*; *Marques, Izquierdo & Pereira, 2013*). Therefore, in order to better analyze the changes in muscular performance of older participants after a strength/power training program, the measurement of movement velocity in tasks like the chair test or the CSQ could have important clinical applications. However, its use among elder populations is still poorly investigated. To the best of our knowledge, this is the first study analyzing mean velocity in the CSQ exercise as a mean to monitor muscular performance in a geriatric population.

Moreover, our results showed a significant correlation between the MMSE cognition test and movement velocity in the CSQ, as well as likely moderate higher scores in the MMSE in the strongest group of participants. Recent research has observed that muscular strength has a positive effect on cognition as, for example, it has been showed that a progressive resistance training program produces a significant improvement in the cognitive function in individuals with mild cognitive impairment (*Mavros et al., 2017*). It seems that some of the main factors that could explain the beneficial effects of resistance training in cognition are the reduction on serum homocysteine and the increased concentrations of insulin-like growth factor I that this type of training produce, which has been associated with impaired cognitive performance or increased neural growth, respectively(*Liu-Ambrose & Donaldson, 2008*). Therefore, our results are consistent with previous research and highlights the relationship between muscular performance and cognition in elders, since there was a significant trend by which participants with higher velocity in the CSQ had better scores in the MMSE cognition test.

However, this study has two main limitations that should be considered when trying to replicate it: first, the average age of the participants was about 72 years and, therefore, conclusions should not be generalized to older geriatric populations. Finally, participants in our study had a mean score in the Berg-scale of 53.3 points, which is considered as "low risk" of falling. Again, conclusions should be taken with precaution as other geriatric populations at higher risks of having falls were not studied in our investigation. These results might help strength and conditioning coaches and physical therapists working with elder populations to monitor muscular performance in a time-efficient, effortless and affordable way within their fall-prevention protocols.

## CONCLUSION

This study showed that the mean velocity recorded by a smartphone app during the CSQ exercise is positively related to measures of functional performance, balance and cognition in elders.

### Funding
The authors received no funding for this work.

### Competing Interests
The authors declare that they have no competing interests.

### Author Contributions
- Carlos Balsalobre-Fernández conceived and designed the experiments, performed the experiments, analyzed the data, contributed reagents/materials/analysis tools, prepared figures and/or tables, authored or reviewed drafts of the paper, approved the final draft.
- Ángel Cordón performed the experiments, contributed reagents/materials/analysis tools, authored or reviewed drafts of the paper, approved the final draft.

- Nazaret Unquiles performed the experiments, contributed reagents/materials/analysis tools, authored or reviewed drafts of the paper, approved the final draft.
- Daniel Muñoz-García conceived and designed the experiments, performed the experiments, analyzed the data, contributed reagents/materials/analysis tools, prepared figures and/or tables, authored or reviewed drafts of the paper, approved the final draft.

## Human Ethics

The following information was supplied relating to ethical approvals (i.e., approving body and any reference numbers):

The study protocol complied with the Declaration of Helsinki for Human Experimentation and was approved by the ethics committee of the institutional review board (CSEULS-PI-059/2015) of La Salle University.

## Data Availability

The raw data are provided in a Supplemental File.

## Supplemental Information

Supplemental information for this article can be found online at http://dx.doi.org/10.7717/peerj.4712#supplemental-information.

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
