# Peer review of "Movement velocity in the chair squat is associated with measures of functional capacity and cognition in elderly people at low risk of fall"

_PeerJ, doi:10.7717/peerj.4712_

## Round 0.1 · original submission · Major Revisions

Your manuscript was evaluated by expert reviewers.

Please address reviewer´s concerns in detail.

Also, take this chance to:

-improve the writing style and clarity of your paper.
-adapt your manuscript to PeerJ policies.
-use clear and unambiguous text that conforms to professional standards.
-include sufficient introduction and background to demonstrate how the work fits into the broader field of knowledge, with relevant prior literature appropriately referenced.
-use the format of ‘standard sections’ from PeerJ Instructions for Authors (a significant modification from PeerJ suggested structure should be made only if this modification significantly improve clarity or conform to a discipline-specific custom).
-please double check that your research question must be clearly defined, must be relevant and meaningful, identifying the knowledge gap being investigated and how the study contributes to filling that gap.
-please double check to provide a Methods section with sufficient information to be reproducible by another investigator.
-review that your data be robust and statistically sound.
-assure that your conclusions be appropriately stated, connected to the original question investigated, and limited to those supported by the results.
-speculation (especially in the discussion of your results) is welcomed, but should be identified as such.


Sincerely,
Rodrigo Ramírez-Campillo
Academic Editor
PeerJ

·

Basic reporting

No comment

Experimental design

No comment

Validity of the findings

No comment

Additional comments

The authors should be commended for presenting a concise and very well-written manuscript. Below I provide specific comments of the different sections of the manuscript that I hope they could contribute to improve the presented manuscript.

ABSTRACT
Line 84. Change “consisting on” by “consisting of”.
Line 87. Delete N=40 because it is redundant.
Line 88. It would be interesting to include the load used in the CSQ test.
Line 89. Delete “Additionally”
Line 92. Use the abbreviation for “chair squat exercise” or remove this abbreviation from the abstract.
Line 93. … correlations “were obtained” between…
Line 95-96. If possible, include the value of the ES after each variable instead of using the range.
Line 97-99. I think that the conclusion would be more informative if the author indicates in what consisted the “muscular performance test” instead of the device that was used for the measurement.

INTRODUCTION
Line 118-119. Introduce the abbreviations of these variables if the abbreviations are going to be used later in the manuscript.
Line 122. Change “this” by “these”.
Line 124-127. Consider to change this sentence for clarity to: “Finally, it has been shown that the velocity recorded against a submaximal load can be used to accurately quantify the maximal strength capacity in a large number of resistance training exercises”. Add at the end of this sentence several references that support this statement.
Line 129. Change “correlations” by “relationship” or “association”.
Line 132. Consisting “of”.
Line 135-136. I miss a couple of sentences in the introduction to justify the hypothesis of the authors.

MATERIAL AND METHODS
General comment: I suggest to describe first the chair squat test and then all the remaining variables that were correlated with the performance in the chair squat test.
Line 138. “Matherial” should be changed to “Material”.
Line 143. Consider to change to: “Participants were also divided…”
Line 149. Change “participant´s” to “participants´”
Line 159. If authors decide to use abbreviations (e.g., HGS), they should use these abbreviations consistently through the manuscript.
Line 164-165. I miss one parenthesis after 0-20.
Line 179. “ms” should be changed to “s”?
Line 186. Change “calculations” by “statistical analyses”.
Line 190-192. This sentence would fit better when describing the chair squat test.
Line 196. Why using 95%CI for correlations and 90%CI for standardized mean differences.

RESULTS
The results section is concise and very easy to follow.

DISCUSSION
Line 222-224. It could be interesting to highlight in the discussion the benefits of the test used in the present study compared to the 30-s chair test (e.g., faster assessment).
Line 234. The word “thus” could be omitted.

CONCLUSIONS
Consider to change to: “This study showed that the mean velocity recorded by a smartphone app during the chair squat exercise is positively related to…”.

·

Basic reporting

I want to thank the author for write this interesting manuscript. However:

1) I suggest count with a professional English servirse to improve the re-phrasing, use of punctuations and minor mistakes with grammar.

2) The reference need to be expanded to give support some topics as methods. Please see attached comments.

3) The manuscript fulfils the scientific structure. But i suggest to add a universal format of data .csv o .txt. Minor changes are attached in comments.

4) The hypothesis is relevant for clinical context, but needs to improve their previous knowledge (introduction), discussion and conclusion with major changes. Please see attached comments.

Experimental design

1) the manuscript fulfils the aims of the journal.

2) the aims of the study requiered minor changes. Please see the comments attached.

3) the manuscript fulfils ethical standard.

4) Method needs major revision. Please see the comments attached.

Validity of the findings

1) If the authors could support the methodological deficits, the manuscript will show an interesting tool for clinician in world-wide.

2) Data needs revision. Please see the comments attached.

3) Conclusion need major revision. Please see the comments attached.

4) Speculation need major revision. Please see the comments attached.

Additional comments

Dear Editor:

1) the manuscript is na original idea but need to major revision in methods, if the article do not present in their next version eligibility criterias' need to be rejected. However could be a mistake of authors.

2) The manuscript is interesting and original. I invite to authors to perform the revision contained in the manuscript to improve the quality of this research.

---

## Round 0.2 · accepted · Accept

Dear authors,

Congratulations for the great work. The expert reviewers are now satisfied with the work done, as well as myself.

The manuscript presents some minor details, which can be improved during the production phase.

Once again, congratulations.

·

Basic reporting

'No comment'

Experimental design

'No comment'

Validity of the findings

'No comment'

Additional comments

The authors accomplished a well scientist manuscript revision. I appreciate the scientific improvement of the manuscript and i think now could be published. Only some minor details must be checked before to publish the manuscript as punctuation in line 119, space of parenthesis in line 140, the correction of 'nor' in line 146 if was a mistake and the parenthesis use in line 157-158.